PLOS · Biology

# Humans have a longer period of cortical maturation across depth and hierarchy than macaques

Monami Nishio ®*, Xingyu Liu, Allyson P. Mackey, Michael J. Arcaro

Department of Psychology, School of Arts and Sciences, University of Pennsylvania, Philadelphia, Pennsylvania, United States of America

* monami@sas.upenn.edu

## Abstract

Postnatal brain development in primates involves prolonged structural maturation of the cortex, laying the foundation for advanced perceptual and cognitive functions. In humans, cortical development is thought to progress along a hierarchical gradient from early-maturing sensorimotor regions to later-developing association areas. However, developmental changes across cortical depth, which contribute to both local computation and large-scale network integration, have been largely unexplored. It also remains unclear which features of these maturational trajectories are conserved across primates and how they may have been further extended or modified in humans. Using the T1-weighted/T2-weighted (T1w/T2w) MRI ratio as a noninvasive measure of cortical microarchitecture, we systematically compared depth-dependent and regional developmental trajectories in humans and macaques. We identified a conserved "inside-out" gradient of maturation, with deeper cortical depths exhibiting steeper increases in T1w/T2w ratio and earlier plateaus than superficial depths. This depth-dependent pattern was embedded within a broader hierarchical gradient of maturation across the cortical surface, extending from sensorimotor regions to association cortex. While the spatial structure of these gradients was shared across species, humans exhibited markedly prolonged development across the entire cortical hierarchy, including both sensory and association cortices, and across cortical depths. These findings suggest that conserved developmental gradients are elaborated in humans to support an extended window of postnatal plasticity, enabling the experience-dependent refinement of cortical circuits that underlie the complex, integrative functions central to human perception and cognition.

## Introduction

Postnatal brain development in primates is marked by prolonged changes in cortical microstructure that support the emergence of complex perception and cognition. In

**Data availability statement:** Both T1w and T2w images of humans are available through the Human Connectome Project–Development (HCP-D; https://www.humanconnectome.org/study/hcp-lifespan-development) and the Human Connectome Project–Young Adult (HCP-YA; https://www.humanconnectome.org/study/hcp-young-adult). Users must register to gain access to the data. Both T1w and T2w images of macaques are available through the UNC-Wisconsin Neurodevelopment Rhesus MRI Database (https://www.nitrc.org/projects/uncuw_macdevmri). All code used for the analyses is available on GitHub at https://github.com/monami-nishio/prolonged_cortical_maturation and on Zenodo at https://zenodo.org/records/16786211 (DOI: 10.5281/zenodo.16786211).

**Funding:** This work was supported by a Whitehall Foundation grant (to M.J.A.), NIH grants P30EY12196, R01EY025670, R01NS123778 (to M.J.A.), and NSF grants 2045095 (to A.P.M.). M.N. was supported by Quad Fellowship and Nakajima Foundation Scholarship. The funders had no role in the study design, data collection and interpretation, or the decision to submit the work for publication.

**Competing interests:** The authors have declared that no competing interests exist.

**Abbreviations:** BBR, boundary-based registration; DMN, Default Mode Network; DUC, Data Use Certification; GAM, Generalized Additive Models; HCP, Human Connectome Project; HPL, Harlow Primate Laboratory; IACUC, Institutional Animal Care and Use Committee; IR, Inversion Recovery; MBP, myelin basic protein; MT, magnetization transfer; MTR, magnetization transfer ratio; MWF, myelin water fraction; NIDCR, National Institute of Dental and Craniofacial Research; NIMH, National Institute of Mental Health; NINDS, National Institute of Neurological Disorders and Stroke.

humans, major white matter tracts are already identifiable at term birth [1], but they remain structurally and functionally immature [2]. Prior to 36 weeks of gestation, unmyelinated white matter predominates, followed by a rapid increase in myelinated fibers during infancy and early childhood that continues more gradually into early adulthood [3]. Myelination enhances the efficiency of neural signaling, supports the regulation of synaptic plasticity, and fosters the emergence of functional specialization [4–7]. These processes are especially pronounced in larger-brained, gyrencephalic primates, where prolonged myelination supports the refinement of distributed cortical networks underlying advanced sensory and cognitive abilities [7]. Notably, myelin accumulation has been linked to the closing of critical periods, the temporally restricted windows of heightened neural plasticity that are essential for functional brain maturation [8,9]. In parallel, cortical maturation involves other microstructural changes, including dendritic arborization [10], synaptic density [11], and glial cell maturation [12], particularly in association regions with prolonged developmental timelines.

The structural maturation of the human brain follows a hierarchical trajectory that aligns with cortical functional organization along the sensorimotor-to-association (S-A) axis [13–17]. Primary sensory and motor regions, associated with rapid, reliable processing, mature early, while association cortices, which support integrative functions across distributed networks, show protracted development [16]. However, the evolutionary origins and continuity of these developmental hierarchies remain unresolved. A previous study in chimpanzees found that the density of myelinated axons reaches its maximum level by adolescence in most cortical regions, with the exception of the frontopolar cortex, which continues to increase gradually beyond adolescence. This suggests a hierarchical maturation pattern similar to that observed in humans [18]. However, this study relied on *ex vivo* histology with limited sampling across the cortical surface, making it difficult to determine global cortical gradients or distinguish shared from human-specific maturational features.

The laminar organization of cortex, characterized by systematic variation in cell types and connectivity patterns, is a core feature of cortical hierarchies [19,20]. This architecture underpins processes ranging from sensory integration to higher-order cognition [19,20]. Despite its crucial role in organizing hierarchical networks, the developmental trajectory of laminar structure and how it shapes functioning across the cortical hierarchy remain poorly understood. Recent studies in rodents suggest that myelination follows an "inside-out" pattern, where deeper cortical layers become myelinated earlier than superficial layers [21–24]. For example, immunohistochemical studies of myelin basic protein (*MBP*) in mice show that the first myelinated axons appear in layers V–VI [21]. Whether this developmental pattern is conserved in primates, particularly given the expanded cortical hierarchy [25] and unique developmental features of the cortical plate [26], remains an open question. In primates, especially humans, the expansion of superficial layer neurons [26] and their increased connectivity complexity [27] support a laminar architecture that is both

structurally differentiated and functionally specialized [28]. Understanding how depth-dependent structural maturation unfolds remains a key question in developmental neuroscience.

Noninvasive imaging approaches, including T1w/T2w ratios and magnetization transfer (MT), have begun to reveal depth-wise variation in cortical structural development in humans. Recent studies have reported steeper increases in signal intensity in middle and deep cortical depths compared to superficial depths from adolescence to adulthood [14,29]. However, the limited age ranges tested make it difficult to determine whether changes reflect concurrent maturation at different rates or true hierarchical variation in developmental timing across cortical layers. Adding further complexity, regional variability in the laminar maturation of cytoarchitecture has been observed in marmosets [30], raising the possibility that the temporal unfolding of cortical depth-dependent maturation may vary across cortical regions in primates.

While cross-species comparisons are critical for identifying conserved principles of cortical development, humans exhibit exceptionally prolonged maturational timelines compared to other primates [18,31]. In macaques, myelination largely stabilizes within the first three years of life [31,32], while our closest living relatives, chimpanzees, exhibit a longer period of development, with myelination continuing to increase into the period of sexual maturity (ages 4–6 years) [18]. In humans, myelination progresses along a markedly extended timeline, continuing into the third decade and beyond [18,33]. This prolonged maturation is thought to allow for extended plasticity, providing a longer window for environmental influences to shape cortical circuits during postnatal development [17,18]. Such a timeline raises the possibility of human-specific modifications to depth-dependent maturation. Superficial layers, with their dense cortico-cortical connections, may show especially prolonged development in humans, consistent with their central roles in integrating information across distributed networks [20,28]. Although much attention has focused on the evolutionary expansion of superficial circuitry, deeper cortical layers may also exhibit prolonged developmental timelines, as their structural maturation supports subcortical pathways essential for sensory processing, attention, and other cognitive functions [20]. Resolving the time course and organization of these developmental patterns across species is essential for understanding how conserved mechanisms of cortical development interact with human-characteristic adaptations.

Advances in neuroimaging allow for *in vivo* comparisons of cortical structural organization across species. The T1w/T2w ratio was introduced by Glasser and colleagues [34] as a proxy for measuring cortical myelin, based on its correspondence with known histological myelination patterns [34]. Supporting this interpretation, postmortem MRI scans from individuals with multiple sclerosis showed lower T1w/T2w ratio values in demyelinated tissue compared to myelinated areas [35]. Importantly, T1w/T2w is not exclusively sensitive to myelin but also reflects a broader set of microstructural features, including dendritic density, glial architecture, and iron content [36]. This composite sensitivity makes the T1w/T2w ratio particularly valuable for capturing regional and depth-dependent patterns of cortical structural maturation across development. T1w/T2w also has been shown to correlate spatially with other myelin-sensitive MRI measures, including the longitudinal relaxation rate (R1) [37], magnetization transfer ratio (MTR) [38,39], and myelin water fraction (MWF) [36]. While these alternative measures may offer somewhat improved specificity to myelin, they require specialized acquisition protocols and are less feasible for large-scale, cross-species developmental studies. T1w/T2w, in contrast, is widely available for use in both human and nonhuman primates [13,40], has been validated across developmental stages [16,41], and is included in several large open-access datasets [42,43], making it ideally suited for cross-species comparative studies [44] on structural development.

In the current study, we developed a cross-primate analysis pipeline to quantify regional and depth-dependent variation in T1w/T2w ratio patterns across development in humans and macaques. This framework enables us to identify both conserved and species-specific patterns of maturational trajectories in cortical structural development. By systematically comparing developmental patterns across cortical regions and depths, we aim to elucidate how shared developmental mechanisms are elaborated to support species-specific sensory and cognitive capabilities in the primate brain.

PLOS Biology

## Results

### Regional and depth-dependent T1w/T2w ratio variability in macaques and humans

To establish a normative reference for subsequent developmental comparisons, we first characterized regional and depth-dependent variation in cortical T1w/T2w ratios in late adolescent macaques (Fig 1A, ages 2–3 years, $N = 25$) and young adult humans (Fig 1B, ages 22–36 years, $N = 1,113$). Cortical surfaces were parcellated using species-specific anatomical templates: the CHARM level 6 template [45,46] for macaques, which includes 139 cortical parcels, and the Schaefer 400 template [47] for humans.

To enable cross-species comparisons, human cortical regions were grouped along the sensorimotor-association (S-A) axis, a primary organizational gradient spanning from sensory and motor cortices to transmodal, heteromodal, and paralimbic association cortices [48–50]. While a similar axis has recently been described in macaques using tracer connectome data [49], this approach lacks full cortical coverage, particularly in association regions, limiting its utility for whole-brain comparisons. To address this limitation, we used geodesic distance along the cortical surface from pre-defined association areas as a proxy for cortical hierarchy in macaques [49]. This measure corresponded with both the tracer-derived S-A axis in macaques (S1A–S1C Fig) and the multimodal-derived S-A axis in humans (S1D–S1F Fig) [42], establishing a framework for direct comparisons of the entire cortical hierarchy between species. Using these hierarchical frameworks, cortical parcels were classified into sensorimotor, middle, and association regions in both species. In both macaques and humans, the T1w/T2w ratios were highest in sensorimotor regions and showed a linear decline along the S-A axis toward association regions (Fig 1C; Macaques $R^2 = 0.096$, $P < 0.001$, Fig 1D; Humans $R^2 = 0.354$, $P < 0.001$). These findings replicate prior work [51] and demonstrate a shared large-scale gradient in cortical structural organization.

To investigate how these hierarchical patterns vary across cortical depth, we segmented the gray matter into six equi-volumetric bins using LayNii [52]. Because anatomical MRI lacks the resolution necessary to precisely delineate histological cortical layers, this binning approach provides a consistent method for sampling depth-wise variation across the cortical surface in both humans and macaques. In both species, T1w/T2w ratios were significantly higher in deeper bins than superficial bins throughout the cortical hierarchy, across sensorimotor, intermediate, and association regions (Fig 1E; Macaques, Fig 1F; Humans, all ANOVAs $F > 9.236$, $P < 0.001$), replicating prior findings [29,53,54].

To assess correspondence with independent histological features, we compared both regional and depth-wise variation in T1w/T2w ratios in adolescent macaques (ages 2–3 years, $N = 25$) to the layer-specific expression of *MBP* from previously published postmortem macaque data [55]. T1w/T2w values showed significant correlation with MBP expression across the cortical surface and depth (see S1 Results and S2 Fig). In young adult humans (ages 18–30 years, $N = 42$) [56], we observed that regional and depth-wise patterns in T1w/T2w corresponded to those derived using R1 mapping (1/T1) [57], a related MRI measure with higher specificity to myelin (see S1 Results and S3 Fig). The significant correlation between T1w/T2w ratio and R1 values across the cortex (S3B Fig) further supports the interpretation that T1w/T2w captures biologically meaningful variation in cortical tissue organization. Together, these results provide a validated framework for cross-species comparisons of T1w/T2w and establish a foundation for cortical maturation across cortical regions and depths.

### Hierarchical development of T1w/T2w ratio along the sensorimotor-association axis is protracted in humans compared to macaques

Having characterized depth-wise structural gradients across the cortical hierarchy in adults, we next examined how these patterns unfold across development. We analyzed regional T1w/T2w trajectories in macaques from 1 month to 3 years ($N = 33$) and in humans from 5.5 to 36 years ($N = 1,730$) (Fig 2A). These age ranges capture overlapping developmental stages, given that macaque brain development proceeds approximately four times faster than human development [58–67]. This 4:1 scaling is derived from developmental alignments of over 270 events, including neurogenesis, axon

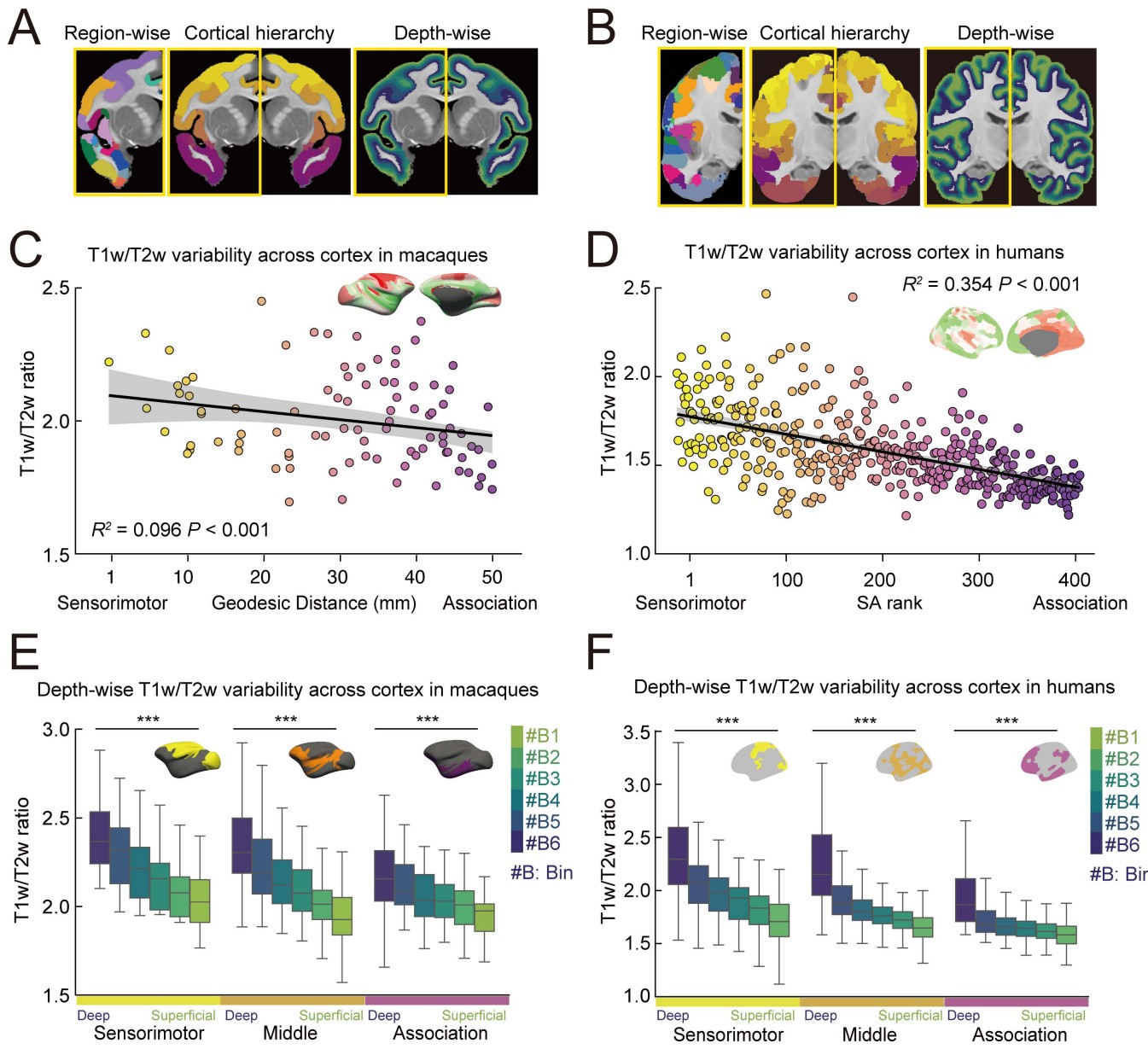

**Fig 1. T1w/T2w ratio across cortical hierarchy and depth in macaques and humans. (A, B)** Schematic of the analytical approach for evaluating the T1w/T2w ratio across cortical regions and depths in macaques (A) and humans (B). The left panel displays the discrete parcels of CHARM level 6 [45,46] for macaques and Schaefer 400 [47] for humans. The middle panel labels the parcels based on geodesic distance for macaques or the sensorimotor-association axis for humans, with colors transitioning from yellow (sensorimotor) to purple (association). The right panel visualizes the laminar organization, with colors transitioning from deep blue (deep layers) to light green (superficial layers). **(C, D)** Distribution of T1w/T2w ratios along the geodesic distance for macaques (C, $R^2 = 0.096$, $P < 0.001$) and along the sensorimotor-association (S-A) axis for humans (D, $R^2 = 0.354$, $P < 0.001$). **(E, F)** Cortical depth-wise T1w/T2w ratios within sensorimotor, middle, and association regions in macaques (E) and humans (F); ANOVA ***$P < 0.001$. The data underlying this figure can be found at https://github.com/monami-nishio/prolonged_cortical_maturation.

outgrowth, connectivity refinement, brain volume growth, and early behavioral milestones, compiled across 19 mammalian species [58,64–67]. We hypothesized that the T1w/T2w trajectories across development in both species would follow the hierarchical gradient observed in adults, with earlier maturation in sensorimotor cortex compared to association regions.

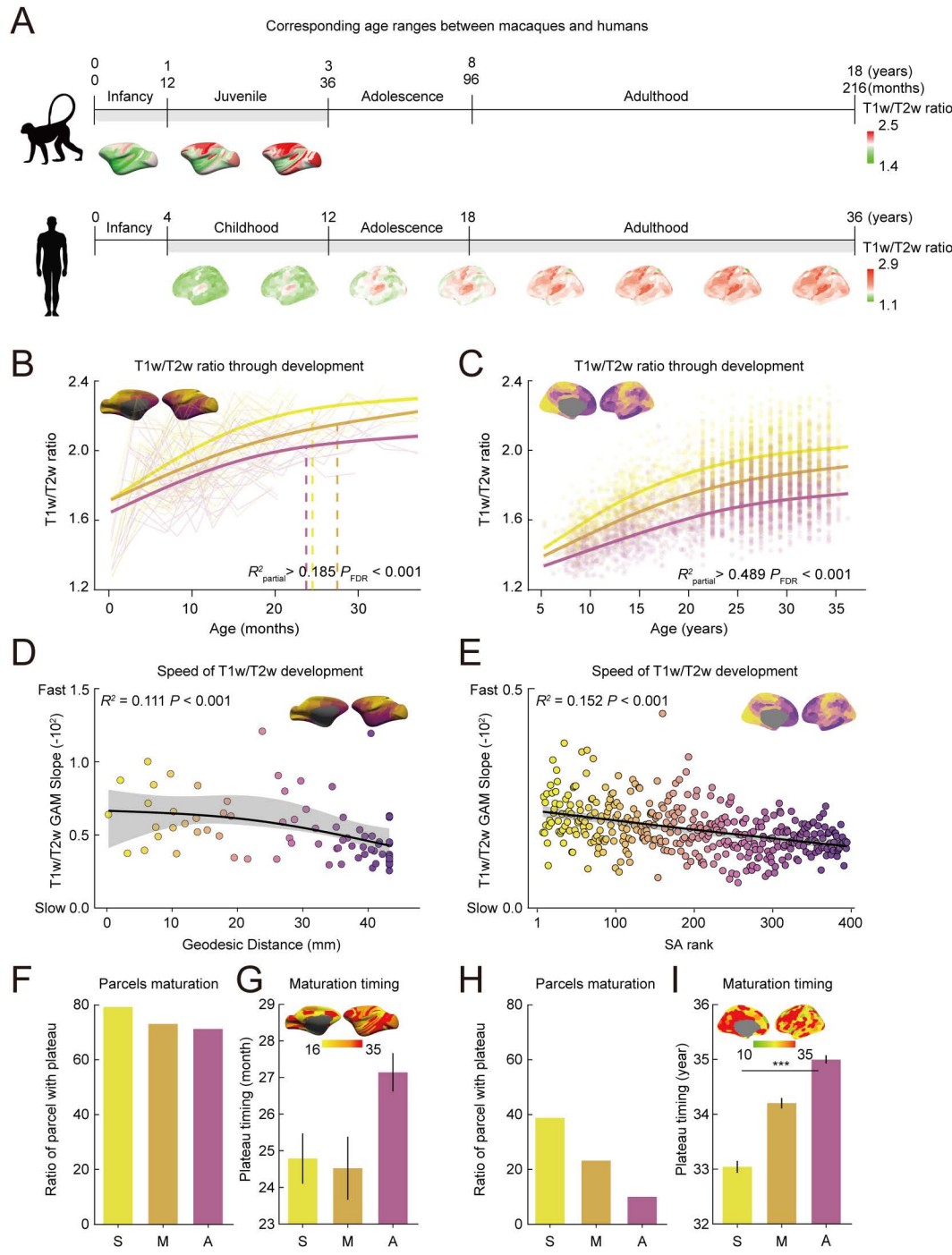

Fig 2. Development of T1w/T2w ratio across cortical hierarchy in macaques and humans. (A) Age ranges for the macaques and humans over-lapped based on the 4:1 age scaling ratio. (B, C) Developmental trajectories of the T1w/T2w ratio for sensorimotor (yellow), middle (orange), and association (purple) cortical areas for macaques (B) and humans (C). Solid lines illustrate Generalized Additive Model (GAM)-predicted fits along with their 95% confidence intervals (B; Sensorimotor; $R^2_{partial} = 0.317$, $P_{FDR} < 0.001$, Middle; $R^2_{partial} = 0.195$, $P_{FDR} < 0.001$, Association; $R^2_{partial} = 0.185$, $P_{FDR} < 0.001$, C; Sensorimotor; $R^2_{partial} = 0.498$, $P_{FDR} < 0.001$, Middle; $R^2_{partial} = 0.539$, $P_{FDR} < 0.001$, Association; $R^2_{partial} = 0.489$, $P_{FDR} < 0.001$). Dashed lines indicate the age at which slope of the T1w/T2w ratio vs. age becomes nonsignificant. (D, E) GAM slopes along the geodesic distance for macaques (D, $R^2 = 0.111$, $P < 0.001$) and along the sensorimotor-association (S-A) axis for humans (E, $R^2 = 0.152$, $P < 0.001$). The GAM-predicted fit is displayed with a 95% confidence interval (D; $R^2 = 0.111$, $P < 0.001$, E; $R^2 = 0.152$, $P < 0.001$). (F, H) Proportion of regions reaching plateau for sensorimotor (S), middle (M), and

association (A) cortex in macaques (F) and humans (H). **(G, I)** Average timing of plateau for cortical regions in sensorimotor (S), middle (M), and association (A) regions for macaques (G; in months, ANOVA $F = 1.353$, $P = 0.262$) and humans (I; in years, ANOVA $F = 14.725$, $P < 0.001$). ANOVA ***$P < 0.001$. The data underlying this figure can be found at https://github.com/monami-nishio/prolonged_cortical_maturation.

We also predicted prolonged development in humans compared to macaques [18]. To test these predictions, we modeled age-related change in each region using Generalized Additive Models (GAM) [68]. The contribution of age was estimated by calculating the differential in $R^2$ values between the full and reduced models omitting age. The direction of developmental change ($R^2_{partial}$) was determined by the sign of the average first derivative of the fitted age function.

In macaques, the T1w/T2w ratio increased significantly with age for all cortical regions, including sensorimotor, middle, and association regions, before plateauing around 24–30 months. This timing corresponds developmentally to approximately 8–10 years of age in humans based on broader metrics of brain maturation [58] (Fig 2B, all $R^2_{partial} > 0.185$, $P_{FDR} < 0.001$). In contrast, human T1w/T2w ratio continued to increase with no detected plateau in any cortical region, even in individuals up to 36 years old, a developmental period far exceeding the corresponding age range analyzed in macaques (Fig 2C, all $R^2_{partial} > 0.489$, $P_{FDR} < 0.001$). These results demonstrate that structural maturation follows a markedly prolonged trajectory in humans compared to macaques.

The rate and timing of development varied across cortical regions in both macaques and humans, reflecting the hierarchical organization of the cortex. In both species, steeper increases in T1w/T2w ratio were observed in sensorimotor regions compared to association regions, with sensorimotor regions plateauing earlier along this gradient (Fig 2D, $R^2 = 0.111$, $P < 0.001$; Fig 2E, $R^2 = 0.152$, $P < 0.001$). This developmental hierarchy was further supported by characterizing the distribution and timing of plateaus across the S-A axis. In both macaques (Fig 2F) and humans (Fig 2H), a greater proportion of sensorimotor regions reached a plateau compared to association regions. The plateau age was earlier in sensorimotor and middle regions than in association regions for both species, although this timing difference reached statistical significance only in humans (macaques: Fig 2G, ANOVA $F = 1.353$, $P = 0.262$; humans: Fig 2I, ANOVA $F = 14.725$, $P < 0.001$). The hierarchical progression of T1w/T2w ratio development along the S-A axis was also evident in humans when cortical hierarchy was estimated using geodesic distance from the default mode network (DMN), paralleling the approach applied in macaques' analysis (S4A–S4C Fig), further supporting the robustness of these findings.

Species differences in developmental timing were most evident when comparing the proportion of cortical regions that reached a plateau in T1w/T2w ratio. By age 3, more than 70% of cortical regions in macaques had plateaued, corresponding to approximately 12 years of human development (Fig 2F). In contrast, by age 36, fewer than half of human sensorimotor cortical regions and only 20% of association regions had reached a plateau (Fig 2H). These findings indicate that the cortical tissue properties indexed by the T1w/T2w ratio, including features linked to microstructural maturation, continue to develop across substantially longer timescales in humans compared to macaques. This extended trajectory exceeds the expected 4:1 cross-species developmental scaling based on broader brain maturation metrics [58,64–67], and is particularly pronounced in association cortex, highlighting the protracted maturation of higher-order cortical systems in humans.

## Structural maturation progresses from deep to superficial cortical depths in macaques and humans

We next examined developmental changes in T1w/T2w ratio across cortical depth to characterize depth-dependent patterns of structural maturation in macaques and humans (Fig 3A; Macaques, Fig 3B; Humans). In both species, deeper portions of cortex exhibited significantly steeper age-related slopes (i.e., increases in T1w/T2w ratio) compared to more superficial portions, consistent across all cortical regions (Fig 3C; Macaques all ANOVAs $F > 7.734$, $P < 0.001$, Fig 3E; Humans all ANOVAs $F > 20.327$, $P < 0.001$). The magnitude of this depth-dependent effect varied across the cortical hierarchy, with the largest differences between superficial and deep portions observed in sensorimotor regions, and smaller differences in association regions (Fig 3D; Macaques $R^2 = 0.061$, $P = 0.014$, Fig 3F; Humans $R^2 = 0.120$, $P < 0.001$).

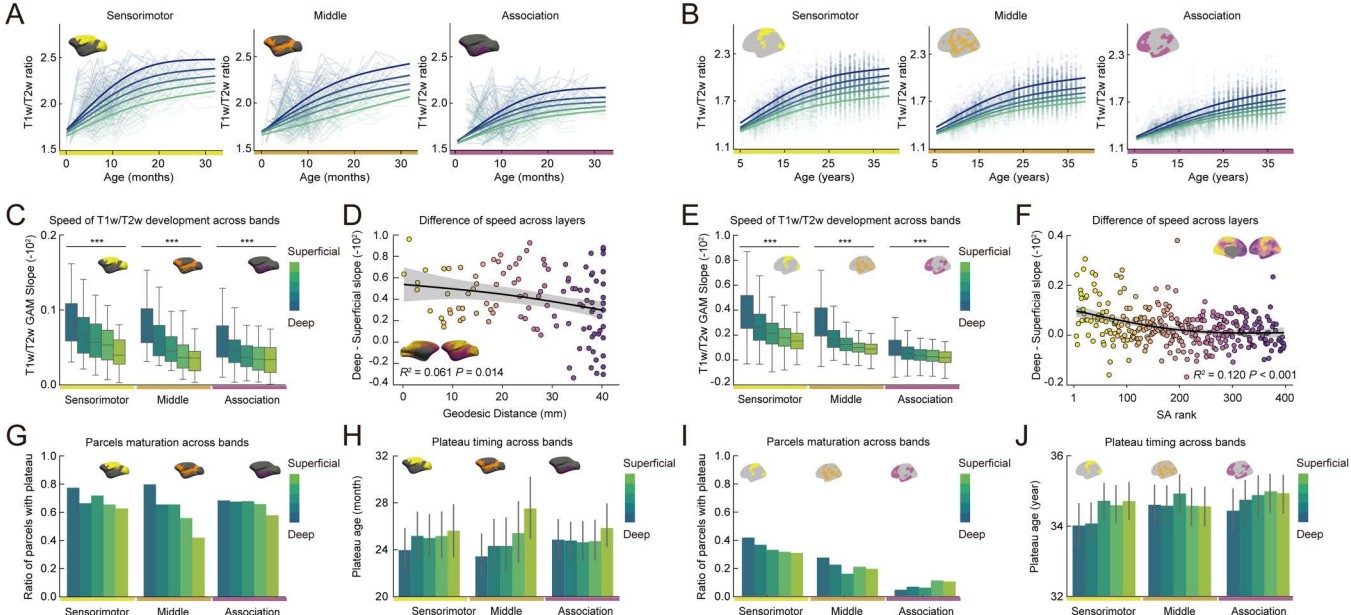

**Fig 3. T1w/T2w ratio across cortical depth in macaques and humans. (A, B)** T1w/T2w ratios and Generalized Additive Model (GAM)-predicted fits for cortical depth bin in sensorimotor, middle, and association regions for macaques (A) and humans (B). **(C, E)** Mean GAM developmental slopes for each cortical depth bin for sensorimotor, middle, and association cortical regions for macaques (C; Sensorimotor; $F = 11.295$, $P < 0.001$, Middle; $F = 11.947$, $P < 0.001$, Association; $F = 7.734$, $P < 0.001$) and humans (E; Sensorimotor; $F = 20.327$, $P < 0.001$, Middle; $F = 55.301$, $P < 0.001$, Association; $F = 99.100$, $P < 0.001$). ***$P < 0.001$. **(D, F)** Slope differences between the deepest and the most superficial bins along the geodesic distance in macaques (D) and along the sensorimotor-association (S-A) axis in humans (F), with GAM-predicted fits and their 95% confidence intervals (D; $R^2 = 0.061$, $P = 0.014$, F; $R^2 = 0.120$, $P < 0.001$). **(G, I)** Proportion of cortical parcels with plateau at each cortical depth bin for sensorimotor, middle, and association regions for macaques (G) and humans (I). **(H, J)** Average timing of plateau in each cortical depth bin for sensorimotor, middle, and association regions for macaques (H; Sensorimotor; $F = 0.355$, $P = 0.840$, Middle; $F = 1.627$, $P = 0.170$, Association; $F = 0.315$, $P = 0.868$) and humans (J; Sensorimotor; $F = 1.666$, $P = 0.156$, Middle; $F = 0.971$, $P = 0.423$, Association; $F = 0.892$, $P = 0.468$). The data underlying this figure can be found at https://github.com/monami-nishio/prolonged_cortical_maturation.

To assess whether depth-dependent variation extended to the timing of developmental stabilization, we next compared the proportion of cortical regions at each depth that reached a plateau in age-related change, as measured by T1w/T2w ratio. In both species, a greater proportion of cortical regions in deeper portions of cortex reached a plateau relative to superficial cortex (Fig 3G; Macaques, Fig 3I; Humans). Although plateaus tended to occur earlier in deeper cortical depths compared to superficial ones, the timing difference was not statistically significant (Fig 3H; Macaques, ANOVAs $F < 1.627$, $P > 0.170$, Fig 3J; Humans ANOVAs $F < 1.666$, $P > 0.156$). Together, these findings reveal a conserved "inside-out" pattern of structural maturation, with earlier stabilization in deeper portions of cortex compared to superficial depths. This depth-dependent trajectory was evident across the cortical hierarchy and consistent in both species. In humans, the same inside-out pattern was observed when cortical hierarchy was estimated using geodesic distance from the DMN, paralleling the approach used in macaques (S4D–S4F Fig), further supporting the robustness of this effect.

Species differences in structural maturational trajectories were evident across cortical depth. In macaques, over 40% of cortical regions reached a plateau by age 3, even in the most superficial portions (Fig 3G). In contrast, in humans, fewer than 45% of cortical regions reached a plateau by age 36, even in the deepest portions (Fig 3I). These results suggest that, relative to macaques, humans retain a prolonged window of structural plasticity across the full depth of the cortical sheet.

## Discussion

This study provides a comprehensive cross-species comparison of cortical structural development, characterizing both depth-dependent and regional trajectories of the T1w/T2w ratio in humans and macaques. We identified a conserved hierarchical gradient of maturation that progresses from primary sensorimotor regions to association cortical areas, alongside an embedded "inside-out" pattern in which deeper portions of the cortical sheet exhibited steeper age-related increases in T1w/T2w ratio and reached developmental plateaus earlier than more superficial portions. Notably, these developmental patterns were present in both species, but were markedly prolonged in humans, extending across cortical regions and depths. This extended developmental window in humans likely reflects species-specific adaptations that build on conserved developmental principles of cortical organization, supporting an expanded period for postnatal circuit refinement and experience-dependent plasticity.

Comparative studies based on histology and transcriptomics, typically limited to select cortical areas, have shown that nonhuman primates, including macaques [32] and chimpanzees [18], reach structural maturation earlier than humans. Our findings confirm and extend these observations using whole-brain neuroimaging across development. In humans, the T1w/T2w ratio continued to increase across the S-A cortical axis well into adulthood, extending far beyond the corresponding developmental window observed in macaques (Fig 4B; sensorimotor regions, Fig 4C; association regions).

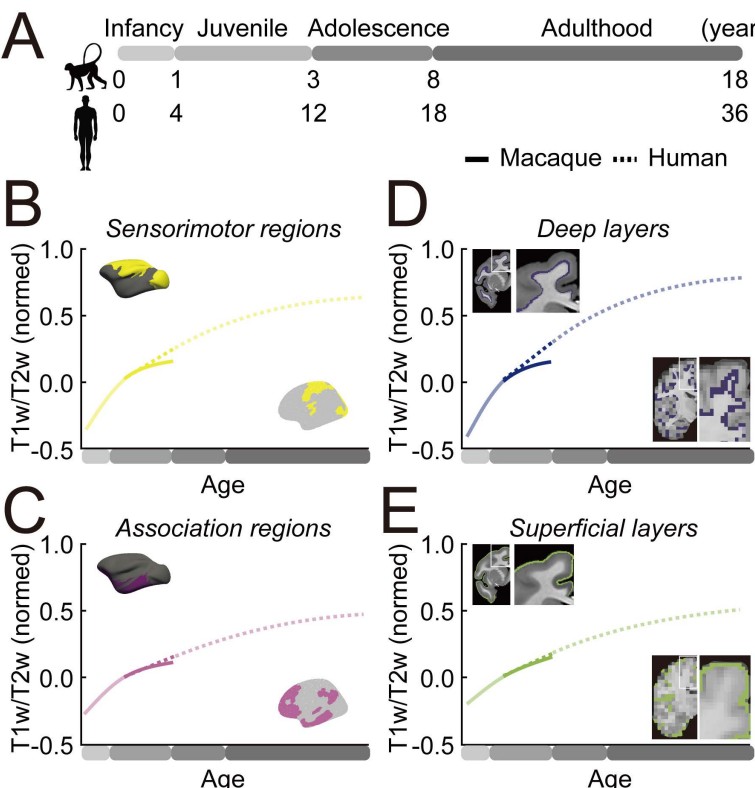

**Fig 4. Summary of regional and cortical depth T1w/T2w Ratio Development in Humans and Macaques. (A)** Age correspondence between macaques and humans assuming a 4:1 scaling. **(B, C)** T1w/T2w ratio developmental trajectories across sensorimotor (B, yellow) and association (C, purple) regions for macaques (straight) and humans (dotted). **(D, E)** T1w/T2w developmental trajectories across deep (D, dark blue) and superficial (E, green) cortical layers for macaques (straight) and humans (dotted). T1w/T2w values are adjusted relative to 1 year old (macaque) and 4 years old (human) as the baseline ($y=0$). The age spans that overlap between macaques and humans are shown in solid colors, while the non-overlapping age span are plotted in diluted colors.

This prolonged maturation was evident not only across cortical regions, but also across cortical depth, with both deep and superficial portions of cortex showing delayed stabilization (Fig 4D; deep layers, Fig 4E; superficial layers). Given that T1w/T2w variation reflects tissue properties associated with myelination and microstructural differentiation, these findings suggest that features relevant to neural signaling and circuit refinement remain developmentally dynamic in humans well into adulthood. This likely includes both cortico-cortical pathways densely represented in superficial cortex and subcortical pathways, including thalamic inputs, that predominantly engage deeper layers. The extended period of structural maturation in humans may support an extended window for experience-dependent plasticity, potentially supporting the development of complex cognitive abilities and flexible behavior across the life span [69]. In line with this interpretation, prior work has demonstrated that intracortical myelination in humans correlates with functional network organization, suggesting that extended structural development could facilitate regionally specialized functions [70]. This view is consistent with neuroconstructivist accounts, which propose that neocortical evolution favors enhanced capacity for learning and adaptability over the early emergence of prespecified core knowledge (i.e., hyper-specialization) [71].

The prolonged cortical development observed in humans is anchored to conserved maturational gradients shared with macaques. Both species displayed a hierarchical trajectory of structural maturation across cortex, progressing from sensorimotor to association regions. This sequence may serve as an evolutionary foundation for regional differentiation in cortical structure. Within this evolutionary context, our results indicate that the human trajectory, while remarkable for its protracted structural maturation, is not extraordinary, as it builds on conserved mechanisms underlying cortical development across primates [72]. In this view, the scalable extension of postnatal development in primates, particularly in association cortices, may support species-specific adaptations such as the emergence of complex perceptual and cognitive functions in humans.

Previous studies indicate that myelination increases more steeply in deeper cortical layers than in superficial layers during adolescence and early adulthood [14,29]. However, mapping the full developmental trajectory of cortical tissue maturation in humans remains challenging, as it involves multiple interwoven processes, including myelination, dendritic arborization, and glial changes, that continue to unfold over several decades. Leveraging the shorter developmental window in macaques, we found that deeper layers not only matured more rapidly, but also reached a plateau earlier than superficial layers. A similar depth-dependent pattern was also observed in humans, although fewer than half of human cortical regions reached a plateau by age 36. These findings suggest that the "inside-out" gradient of cortical maturation is a conserved feature of primate cortical development, though it unfolds over a much longer timeline in humans. The functional implications of this pattern likely vary with cortical depth. Earlier stabilization of deeper layers, which contain subcortical projection neurons, may support essential sensory processing and motor control during early postnatal development. In contrast, prolonged maturation of superficial layers, predominantly containing cortico-cortical connections, may sustain a flexible period for postnatal learning and integrative processing across distributed networks [71].

Several potential limitations of this work warrant consideration. Our analyses rely on T1w/T2w ratio as an indirect measure of myelination and other microstructural maturation. While this metric shows correspondence with myelin content in humans, its specificity in nonhuman primates [41] and its sensitivity across cortical depth [14] remain areas of active investigation. Notably, we observed greater variability in macaques, particularly in the relationship between T1w/T2w ratio and cortical hierarchy, as well as in the variance explained by age. This may reflect true biological differences between species, but could also stem from technical factors such as smaller brain size, cortical folding patterns, and signal distortions near air/tissue boundaries, which may disproportionately affect superficial estimates and inflate apparent effects in deeper layers. These effects can be particularly pronounced in medial frontal and inferior temporal regions, especially near the sinuses and ear canals. Although we did not observe widespread segmentation anomalies in our macaque data, future studies could benefit from spin-echo-based imaging protocols that reduce susceptibility to dephasing and signal loss [34]. Another methodological consideration concerns our approach to sampling cortical depth. We used an equivolume algorithm to divide the cortical gray matter into depth bins, consistent with established methods [52,73–75]. While

this approach accounts for cortical curvature and ensures equal volumetric sampling, it does not capture regional variability in laminar structure. Emerging techniques now offer MRI-based whole-brain laminar thickness maps in adult humans [76]. Future work could benefit from biologically-informed laminar maps grounded in histological references such as the Von Economo atlas [77], extended across species and developmental stages.

Taken together, our findings revealed both shared and human-characteristic features of cortical maturation, offering broader insights into developmental and evolutionary organization of the human brain. These results highlight the importance of comparative approaches for distinguishing between aspects of brain development that are inherited from shared features and those that represent species-specific adaptations. In particular, the prolonged structural maturation observed in humans, especially within superficial cortical layers associated with cortico-cortical communication, may provide an extended window of plasticity in association cortices involved in complex behaviors such as abstract reasoning and social cognition. Importantly, this protracted development is not restricted to higher-order regions. Rather, it spans the entire cortical surface, including primary sensory and motor areas. The extended developmental timeline observed in humans surpasses the 4-fold macaque-to-human scaling factor commonly used to align developmental milestones [58]. This widespread prolongation likely reflects adaptations across the cortical hierarchy that support the emergence of advanced perceptual and cognitive capabilities characteristic of human behaviors.

## Materials and methods

### Ethical statement

The research protocol for UNC-Wisconsin Neurodevelopmental Rhesus MRI Database was approved by the Institutional Animal Care and Use Committee (IACUC). Care and treatment of the animals at HPL are designed to meet and exceed the guidelines promulgated by the National Institutes of Health Guide for the Care and Use of Laboratory Animals. The quality of the research findings is predicated on the high quality of care. The research protocol for HCP and HCPD is conducted in accordance with the principles expressed in the Declaration of Helsinki and is approved by the Institutional Review Board at Washington University in St. Louis (HCP-D IRB ID#: 201603135; HCP-YA IRB ID#: 201204036). HCP-YA data were obtained from the Human Connectome Project (HCP) database (https://ida.loni.usc.edu/login.jsp) by creating an account on ConnectomeDB and agreeing to the Open Access Data Use Terms. HCP-D data were obtained from the National Institute of Mental Health (NIMH) Data Archive (NDA, Collection #2846) under an approved NDA Data Use Certification (DUC). Use of HCP and HCP-D data in the present study was approved by the Institutional Review Board at the University of Pennsylvania (IRB ID#: 825656).

### Participants

**Humans.** We analyzed neuroimaging data from 1,730 healthy participants: 617 aged 5.58–21.92 years (54.8% female) from the HCP in Development (HCP-D; Release 2.0) [16], and 1,113 participants aged 22–36 years (54.4% female) from the HCP (S1200 release) [78]. We excluded 13 HCP-D participants due to missing T1w scans or anatomical anomalies, resulting in a final sample of 617 youth. Age distributions are shown in S5A Fig. To test the robustness of our results against sample distribution, we replicated Fig 2C after excluding participants younger than 9 years (due to low sample size) and selecting 50 samples from each 2-year age group to ensure a more balanced distribution across the age range (S5D Fig).

**Macaques.** We analyzed neuroimaging data from 33 rhesus monkeys (Macaca mulatta, 41.2% female) from the UNC-Wisconsin Rhesus Macaque Neurodevelopment Database [44]. Monkeys were reared and housed at the Harlow Primate Laboratory (HPL) at the University of Wisconsin-Madison. Each monkey underwent longitudinal scanning (five time points, except one monkey with four), with intervals based on age at first scan. Age distributions are presented in S5B and S5C Fig. To test the robustness of our results against sample distribution, we replicated Fig 2B by selecting 12 samples from each 6-month age group to achieve a more balanced distribution across the age range (S5E Fig).

For single-nucleus RNA sequencing, we analyzed data from two monkeys as published in Chen and colleagues [55]. Left hemispheres were collected from two male cynomolgus monkeys (*M. fascicularis*; #1, 6-year-old, 4.2 kg; #2, 4-year-old, 3.7 kg).

## Image acquisition

**Humans.** High-resolution T1w MRI images were acquired on a 3T Siemens Prisma with a 32 channel head coil using a 3D multiecho MPRAGE sequence [79,80] (0.8-mm isotropic voxels, TR/TI = 2,500/1,000 ms, TE = 1.8/3.6/5.4/7.2 ms, flip angle = 8°, in-plane (iPAT) acceleration factor of 2, TA = 8:22, up to 30 reacquired TRs). Structural T2w images were acquired using the variable-flip-angle turbo-spin-echo 3D SPACE sequence [81] (0.8 mm isotropic voxels, TR/TE = 3,200/564 ms; same in-plane acceleration, TA = 6:35, up to 25 reacquired TRs). Both pre-scan normalized and nonnormalized reconstructions were generated, with nonnormalized versions used for subsequent processing [82] and both were used to estimate the B–receive field for motion correction between T1w and T2w images. Only the first two T1w echoes were used due to artifacts in later echoes [83]. Volumetric navigators (vNavs) were embedded in T1w and T2w sequences for prospective motion correction and selective k-space reacquisition [84], reducing bias in brain morphometry analyses [85,86]. Low-quality scans were reacquired, and only the highest quality T1w/T2w ratio pair per session was used. Additional 2-mm isotropic gradient echo and spin echo images were acquired for pseudo-transmit field computation [87]. For detailed protocol information, see Harms and colleagues [42].

**Macaques.** High-resolution 3D T1-weighted images were acquired on a GE MR750 3.0T scanner (General Electric Medical, Milwaukee WI) with an 8-channel brain array coil using an axial Inversion Recovery (IR) prepared fast gradient echo (fGRE) sequence (GE BRAVO) (0.55 × 0.55 × 0.8 mm, TI = 450 ms, TR = 8.684 ms, TE = 3.652 ms, FOV = 140 × 140 mm, flip angle = 12°, matrix = 256 × 256, thickness = 0.8 mm, gap = −0.4 mm, 80 percent field-of-view in phase encoding direction, bandwidth = 31.25 kHz, 2 averages). Structural T2-weighted images were acquired using a sagittal 3D CUBE FSE sequence (0.6 mm isotropic voxels, TR = 2,500 ms, TE = 87 ms, FOV = 154 × 154 mm, flip angle = 90°, matrix = 256 × 256, 90% field of view in the phase encoding direction, slice thickness = 0.6 mm, gap = 0 mm, bandwidth = 62.5 kHz, ARC parallel imaging with a factor of 2 acceleration in both phase encoding and slice encoding directions). Animals were scanned under anesthesia. Subjects younger than 6 months were immobilized using inhalant isoflurane, while older subjects received ketamine hydrochloride (10 mg/kg I.M.) and dexdomitor (0.01 mg/kg I.M.). Heart rate and oxygen saturation were monitored throughout the experiment.

## Image preprocessing

**Humans.** Preprocessed structural MRI data were provided as part of the Lifespan HCP Release 2.0 through the NDA (https://nda.nih.gov/). The data were processed using HCP Pipelines (version 4.0.1) within the QuNex container environment (qunex.yale.edu). As part of this pipeline, T1w and T2w volumes were processed through the *PreFreeSurfer* pipeline, which included gradient nonlinearity distortion correction using the N4 [88] method. This correction addresses low-frequency intensity inhomogeneities associated in part with B1$^+$ field variation and improves spatial uniformity in the ratio map. While this does not substitute for explicit B1$^+$ mapping, it does help mitigate one of the key sources of nonbiological variation in ratio-based contrasts. After the distortion correction, initial brain-extraction, and rigid registration into an anterior/posterior-commissure aligned "native" space were performed. Each subject's T2w volume was registered to their T1w volume using boundary-based registration (BBR) [89], and the receiver coil bias field was corrected based on the smoothed square root of the product of the T1w and T2w images. Images were then registered to MNI template space [90]. The *FreeSurfer* pipeline (v6.0.0) [91], optimized for use with high-spatial resolution (>1 mm isotropic) images, was used to compute the "white" and "pial" surfaces, including use of the T2w volume to optimize the pial surface placement. The values for the corpus callosum and ventricles were extracted from individual T1w and T2w images and calibrated using the group-averaged values of all subjects in the HCP dataset [38].

**Macaques.** Preprocessed structural MRI data were provided as part of the UNC-Wisconsin Rhesus Macaque Neurodevelopment Database [44]. AutoSeg [92] was used to perform the processing of the structural images. Bias field corrections were applied using the N4 [88] method for both T1 and T2 images [88]. T1 images were aligned to an external T1 atlas (Emory-UNC atlas at 12 months) using BRAINS fit's rigid body, normalized mutual information registration. T2 images were aligned to the atlas-registered T1 images. These structural images were resampled in atlas space to $0.2375 \times 0.2375 \times 0.2375$ mm resolution. Tissue segmentation was performed using the Atlas-Based Classification for white matter, gray matter, cerebrospinal fluid, and background [93]. Tissue segmentation results were used to generate binary brain masks that were corrected manually by an expert. Both T1w and T2w images were then warped to the NMT template space [45,94]. As was done for the human preprocessing, the values for the pons and ventricles were extracted from individual T1w and T2w images and calibrated using the group-averaged values of all subjects [38].

### T1w/T2w ratio calculation

T1w/T2w maps were generated by dividing the T1w image by the T2w image in standard template space (MNI for humans; NMT for macaques). This division cancels the signal intensity bias related to the sensitivity profile of the radio frequency receiver coils. The T1w/T2w ratio also enhances contrast associated with myelin content, as both images exhibit myelin-related contrast [34] that is inverted in the T2w image relative to the T1w image. Individual T1w/T2w maps were parcellated using the Schaefer 400 parcellation [47] for humans and the CHARM level 6 139 parcellation for macaques [45,46] to calculate the average T1w/T2w value for all vertices within each parcel. These T1w/T2w units are arbitrary, representing relative measures of intracortical myelin content that depend on the scanner field strength and sequence parameters, and should therefore not be directly compared across studies without appropriate normalization.

### Layer segmentation

The cortical gray matter was segmented using the LN2_LAYERS functions from LayNii [52]. The gray matter mask in template space (MNI for humans; NMT for macaques) was segmented according to the equi-volume principle [75], which considers that outer layers have thinner volumes and produces depth bins of equal volume [52,73–75]. For Figs 1 and S2, the cortical gray matter was segmented into six equi-volume bins to maintain correspondence with the biological six layers. For Figs 2 and 3, it was segmented into seven bins, with the first and last bins removed to avoid contamination from white matter or external brain structures, resulting in five bins.

### Geodesic distance from default mode network regions

**Humans.** The geodesic distance along the cortical surface was calculated using tvb-gdist [95] module (https://github.com/the-virtual-brain/tvb-gdist) that approximates the shortest path between two nodes on a triangular surface mesh. We selected parcels in the Schaefer 400 parcellation [47] corresponding to the Default Mode Network (DMN) B from Yeo's 17 networks [96]. Cortical nodes within these DMN parcels were clustered using k-means clustering, with 10% of the clusters randomly selected as seed nodes for efficiency. Each cortical node was then assigned a distance value based on the minimum geodesic distance along the "midthickness" surface to any of the seed nodes. Finally, the average of the minimum geodesic distances for nodes within each cortical parcel of the Schaefer 400 parcellation was calculated.

**Macaques.** The seed nodes were selected from five peak nodes from the principal gradient [49], derived from axonal tract-tracing decomposition, located within clusters corresponding to independent regions of the DMN [49]. The same method for calculating geodesic distance in human data was applied. The average of the minimum geodesic distances for nodes within each cortical parcel of the CHARM level 6 parcellation [45,46] was calculated.

## Alignment with the sensorimotor-association (S-A) axis

**Humans.** We used the S-A axis derived by Sydnor and colleagues [48]. This map integrates various cortical hierarchies, including functional connectivity gradients, evolutionary cortical expansion patterns, anatomical ratios, allometric scaling, brain metabolism measures, perfusion indices, gene expression patterns, primary modes of brain function, cytoarchitectural similarity gradients, and cortical thickness.

**Macaques.** We used the S-A axis derived by Margulies and colleagues [49]. This map represents the principal gradient revealed through the decomposition of the axonal tract-tracing connectivity database CoCoMac [97,98]. To calculate the correlation between geodesic distance and the S-A axis in macaques, we aligned the CHARM level 6 parcellation [45,46] with the Bonin–Bailey parcellation [99,100] on Yerkes19 template [101]. We identified the most overlapping parcel in the Bonin-Bailey parcellation for each CHARM parcel. Parcels in the CHARM parcellation that did not overlap with any parcels in the Bonin-Bailey parcellation were left unassigned.

## Generalized Additive Models

To model both linear and nonlinear relationships between the T1w/T2w ratio and age, we employed GAMs implemented with the mgcv package in R [102]. In these models, the region-averaged T1w/T2w ratio served as the dependent variable, with age modeled as a smooth term, and gender included as a linear covariate. Models were fitted separately for each cortical parcel, using thin plate regression splines as the basis set for the smooth term and the restricted maximum likelihood approach for selecting the smoothing parameter. The smooth term for age generated a spline representing each region's developmental trajectory, with a maximum basis complexity ($k$) set to 3 to avoid overfitting.

For each regional GAM, we tested the significance of the association between the T1w/T2w ratio and age using an analysis of variance (ANOVA), comparing the full GAM model to a nested model without the age term. A significant result, as evaluated by the chi-squared test statistic, indicated that the inclusion of the age smooth term significantly reduced the residual deviance. Using the gratia package in R, we identified the specific age range(s) for each regional GAM where the T1w/T2w ratio showed significant changes by examining the first derivative of the age smooth function (Δ T1w/T2w ratio/Δ age) and determining when its simultaneous 95% confidence interval did not include zero (two-sided).

To quantify the magnitude and direction of the association between the T1w/T2w ratio and age, referred to as a region's overall age effect, we measured effect size by calculating the partial $R^2$ comparing the full GAM model with a reduced model that excluded the age term. We signed the partial $R^2$ based on the average first derivative of the smooth function to indicate the effect's direction. We sorted the parcels based on their S-A rank and divided them into three groups: sensorimotor, middle, and association regions, each consisting of 133 parcels. We then performed the same GAM analysis on the average values across parcels within each group. For the macaque study, given the longitudinal design with multiple measurements obtained from the same monkeys over time, we accounted for within-subject correlation by including subject-specific random effects. This approach controls for variability arising from repeated measurements within the same individuals.

## Alignment with MBP expression

*MBP* expression for each brain region and each cortical layer, as well as the regional averages, was provided by Chen and colleagues [55]. T1w/T2w ratio images of adolescent macaques (ages 2–3 years, $N = 25$) were segmented into six equi-volume bins to correspond with the six biological cortical layers. Using the layer-averaged *MBP* expression values, we calculated the correlation between *MBP* expression and T1w/T2w ratio across regions. Additionally, using layer-specific *MBP* expression data, we calculated the linear correlation between *MBP* expression and T1w/T2w ratio across the six layers for each brain region individually.

## Alignment with R1(1/T1) mapping

R1 (1/T1) maps spanning the entire adult life span (ages 18–80, divided into six age groups: 18–30, 31–40, 41–50, 51–60, 61–70, and 71–80) were provided by Alkemade and colleagues [56]. For this study, we used data from 42 individuals (27 female) in the 18–30 age group. Each individual R1 map was registered to the AHEAD template [56] using ANTs, applying a sequence of rigid, affine, and nonlinear transformations. The Schaefer 400 parcellation and the equi-volumetric six-layer model were also registered to the AHEAD template and used to parcellate each individual's R1 map in AHEAD space. Using parcel-wise R1 maps, we then computed the correlation between the R1 values and the T1w/T2w ratio across brain regions.

## Supporting information

**S1 Results.** Validation of T1w/T2w ratio: *MBP* Expression in Macaques and R1 mapping in humans. (DOCX)

**S1 Fig.** Correlation between geodesic distance and the sensorimotor-association axis. (A, D) Sensorimotor-association axis in macaques (A) and humans (D). **(B, E)** Geodesic distance from association regions in macaques (B) and humans (E). **(C, F)** Correlation between S-A axis and geodesic distance in macaques (C, $r = 0.589$, $P < 0.001$) and humans (F, $r = 0.683$, $P < 0.001$). The data underlying this figure can be found at https://github.com/monami-nishio/prolonged_cortical_maturation. (TIF)

**S2 Fig. Correlation between *MBP* and T1w/T2w ratio across layers in macaques. (A)** *MBP* expression across the cortex. **(B)** *MBP* expression along the geodesic distance from association regions. Brain regions are arranged on a spectrum from farther (yellow) to closer (purple) to the association centers. The GAM-predicted fit is displayed with a 95% confidence interval ($R^2 = 0.250$, $P < 0.001$). **(C)** Correlation between *MBP* expression and the T1w/T2w ratio across the cortex ($r = 0.406$, $P < 0.001$). The regression line is displayed with 95% confidence intervals. **(D)** Schematic illustration of *MBP* expression across cortical layers. **(E)** *MBP* expression in each layer for sensorimotor (yellow), middle (orange), and association (purple) regions. ANOVA $^{***}P < 0.001$. **(F)** Correlation between *MBP* expression and T1w/T2w ratio across layers or bins in representative brain regions. The data underlying this figure can be found at https://github.com/monami-nishio/prolonged_cortical_maturation. (TIF)

**S3 Fig. R1 across cortical hierarchy and depth in humans. (A)** Distribution of R1 along the sensorimotor-association (S-A) axis for humans ($R^2 = 0.189$, $P < 0.001$). **(B)** Correlation between T1w/T2w ratio and R1 across the cortex in humans ($r = 0.499$, $P < 0.001$). The regression line is displayed with a 95% confidence interval ($R^2 = 0.209$, $P < 0.001$). **(C)** Cortical depth-wise R1 within sensorimotor, middle, and association regions in humans (ANOVA $^{***}P < 0.001$). The data underlying this figure can be found at https://github.com/monami-nishio/prolonged_cortical_maturation. (TIF)

**S4 Fig. Development of T1w/T2w ratio along geodesic distance in humans. (A)** T1w/T2w ratio across the cortex in humans. Brain parcels are aligned along geodesic distance from default mode network regions ($R^2 = 0.150$, $P < 0.001$). **(B)** Developmental trajectories of the T1w/T2w ratio for sensorimotor (yellow), middle (orange), and association (purple) cortical areas. Solid lines illustrate Generalized Additive Model (GAM)-predicted fits along with their 95% confidence intervals (Sensorimotor; $R^2_{partial} = 0.466$, $P_{FDR} < 0.001$, Middle; $R^2_{partial} = 0.430$, $P_{FDR} < 0.001$, Association; $R^2_{partial} = 0.356$, $P_{FDR} < 0.001$). **(C)** GAM slope along the geodesic distance for humans. The GAM-predicted fit is displayed with a 95% confidence interval ($R^2 = 0.054$, $P < 0.001$). **(D)** Slope differences between the deepest and the most superficial bins along the geodesic distance in humans, with GAM-predicted fits and their 95% confidence intervals ($R^2 = 0.016$, $P = 0.013$). **(E)** Proportion of

cortical parcels with plateau at each cortical depth bin for sensorimotor, middle, and association regions for humans. **(F)** Average timing of plateau in each cortical depth bin for sensorimotor, middle, and association regions for humans. The data underlying this figure can be found at https://github.com/monami-nishio/prolonged_cortical_maturation. (TIF)

**S5 Fig. Age distribution of human and macaque subjects. (A)** Age distribution of human participants from the HCP-D and HCP datasets. **(B, C)** Age distribution (B) and scan time points per subject (C) for macaque subjects from the UNC-Wisconsin Rhesus Macaque Neurodevelopment Database. **(D, E)** Developmental trajectories of the T1w/T2w ratio in sensorimotor (yellow), middle (orange), and association (purple) cortical areas for humans (D) and macaques (E), based on subsampled data. The data underlying this figure can be found at https://github.com/monami-nishio/prolonged_cortical_maturation. (TIF)

## Acknowledgments

Human Connectome Project (HCP; Principal Investigators: Bruce Rosen, MD, PhD, Arthur W. Toga, PhD., Van J. Weeden, MD) funding was provided by the National Institute of Dental and Craniofacial Research (NIDCR), the National Institute of Mental Health (NIMH), and the National Institute of Neurological Disorders and Stroke (NINDS). HCP data are disseminated by the Laboratory of Neuro Imaging at the University of Southern California. Data used in the preparation of this manuscript were obtained from the National Institute of Mental Health (NIMH) Data Archive (NDA). This manuscript reflects the views of the authors and may not reflect the opinions or views of the NIH or of the Submitters submitting original data to NDA.

*Declaration of generative AI and AI-assisted technologies:* during the preparation of this work, the author used ChatGPT 3.5 to draft the first version of the manuscript. After using this tool, all authors edited the content substantially and take full responsibility for the content of the publication.

## Author contributions

**Conceptualization:** Monami Nishio, Xingyu Liu, Michael J. Arcaro.

**Data curation:** Monami Nishio, Xingyu Liu.

**Formal analysis:** Monami Nishio, Xingyu Liu.

**Funding acquisition:** Allyson P. Mackey, Michael J. Arcaro.

**Investigation:** Monami Nishio, Xingyu Liu, Michael J. Arcaro.

**Methodology:** Monami Nishio, Xingyu Liu, Michael J. Arcaro.

**Project administration:** Michael J. Arcaro.

**Resources:** Michael J. Arcaro.

**Supervision:** Allyson P. Mackey, Michael J. Arcaro.

**Visualization:** Monami Nishio, Xingyu Liu.

**Writing – original draft:** Monami Nishio.

**Writing – review & editing:** Xingyu Liu, Allyson P. Mackey, Michael J Arcaro.

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
