## [Editor Report · Decision Letter 0]

24 Jul 2025

Dear Dr Nishio,

Thank you for submitting your manuscript entitled "Prolonged cortical maturation across depth and hierarchy in humans" for consideration as a Research Article by PLOS Biology.

Your manuscript has now been evaluated by the PLOS Biology editorial staff as well as by an academic editor with relevant expertise and I am writing to let you know that we would like to send your submission out for external peer review.

Once your full submission is complete, your paper will undergo a series of checks in preparation for peer review. After your manuscript has passed the checks it will be sent out for review. To provide the metadata for your submission, please Login to Editorial Manager (https://www.editorialmanager.com/pbiology) within two working days, i.e. by Jul 28 2025 11:59PM.

Kind regards,

Luke

Lucas Smith, Ph.D.

Senior Editor

PLOS Biology

lsmith@plos.org

---

## [Editor Report · Decision Letter 1]

8 Aug 2025

Dear Dr Nishio,

Thank you for your patience while your manuscript "Prolonged cortical maturation across depth and hierarchy in humans" was assessed at PLOS Biology. Your manuscript has been assessed as a revision, in response to a previous round of reviews from another journal, under our portable peer review policy. We were able to confirm the review history with the previous journal, and your response to reviewers and manuscript was assessed by the Academic Editor, who has commented that you have done a good job of responding comprehensively to the reviewers' comments.

Based on our Academic Editor's assessment of your revision, we are likely to accept this manuscript for publication. However, before we can accept your study, we need you to address a few last data and other policy-related requests in another short revision. These are detailed below.

**IMPORTANT: Please address the following editorial requests:

1) TITLE: We would like to suggest a tweak to the title to make it clearer that the study compares human and macaque development. We suggest you change it to something like:

"Humans have a longer period of cortical maturation across depth and hierarchy than macaques"

2) ARTICLE TYPE: Please note that we think your study is closest in scope to our 'short report' format, and we ask that you change the article type in our editorial manager system accordingly. This will not require any changes to the actual manuscript given that your study already fits within our 4 figure limit.

3) ETHICS STATEMENT: Please provide an ethics statement related to the work conducted in this study. A complete ethics statement should include:

-- The full name of the IACUC/ethics committee that reviewed and approved the animal care and use protocol/permit/project license. Please also include an approval number.

-- The specific national or international regulations/guidelines to which your animal care and use protocol adhered. Please note that institutional or accreditation organization guidelines (such as AAALAC) do not meet this requirement.

-- Information about the form of consent (written/oral) given for research involving human participants. All research involving human participants must have been approved by the authors' Institutional Review Board (IRB) or an equivalent committee, and must have been conducted according to the principles expressed in the Declaration of Helsinki.

For data generated by a third party, you can refer to their ethical approval and whether the data was collected as described above. However, my understanding is that for the use of these public datasets (especially the HCP dataset) you would also need to consult with your local IRB and Ethics Committee to determine if it needs to be approved or to have this work declared exempt. If your study was approved by a local ethics committed please include those details (the name of the approval body, and approval number). If it was declared exempt, please also include a note to that effect (and indicate which name of the committee made this declaration).

4) DATA AVAILABILITY: I see that your data availability statement currently says 'All code will be available on GitHub, and the processed data required to replicate the figures will be accessible through BALSA.'

>>please provide the link to your github deposition and the access details for the data deposited on BALSA so that I can check that these meet our data sharing requirements

>>please add a note to each figure legend pointing readers to where they can find the underlying data

>>for code, please note that we cannot accept sole deposition of code in GitHub, as this could be changed after publication. However, you can archive this version of your publicly available GitHub code to Zenodo. Once you do this, it will generate a DOI number, which you will need to provide in the Data Accessibility Statement (you are welcome to also provide the GitHub access information). See the process for doing this here: https://docs.github.com/en/repositories/archiving-a-github-repository/referencing-and-citing-content

>> We understand that you have used third party data for this study and that you may not be able to share the raw data analyzed here (other than the numerical values underlying your figures). That is an acceptable restriction, based on our data availability policy, but we ask that you update your data availability statement to also include a description of the data set and the third-party sources. Please also provide all necessary contact information (ex a website) that others would need to apply to gain access to the data

For more information on our data policy, see here: http://journals.plos.org/plosbiology/s/data-availability

We expect to receive your revised manuscript within two weeks.

*Published Peer Review History*

*Press*

Sincerely,

Luke

Lucas Smith, Ph.D.

Senior Editor

lsmith@plos.org

PLOS Biology

---

## [Editor Report · Decision Letter 2]

25 Aug 2025

Dear Dr Nishio,

Thank you for the submission of your revised Short Report "Humans have a longer period of cortical maturation across depth and hierarchy than macaques" for publication in PLOS Biology, and thank you for addressing our last editorial requests in this revision. On behalf of my colleagues and the Academic Editor, Henry Kennedy, I am pleased to say that we can in principle accept your manuscript for publication, provided you address any remaining formatting and reporting issues. These will be detailed in an email you should receive within 2-3 business days from our colleagues in the journal operations team; no action is required from you until then. Please note that we will not be able to formally accept your manuscript and schedule it for publication until you have completed any requested changes.

PRESS

We frequently collaborate with press offices. If your institution or institutions have a press office, please notify them about your upcoming paper at this point, to enable them to help maximize its impact. If the press office is planning to promote your findings, we would be grateful if they could coordinate with biologypress@plos.org. If you have previously opted in to the early version process, we ask that you notify us immediately of any press plans so that we may opt out on your behalf.

Sincerely, 

Luke

Lucas Smith, Ph.D.

Senior Editor

PLOS Biology

lsmith@plos.org